# Short- and Long-Term Safety and Efficacy of Self-Expandable Leo Stents Used Alone or with Coiling for Ruptured and Unruptured Intracranial Aneurysms: A Retrospective Observational Study

**DOI:** 10.3390/jcm10194541

**Published:** 2021-09-30

**Authors:** François Lebeaupin, Pierre-Olivier Comby, Marc Lenfant, Pierre Thouant, Brivaël Lemogne, Kévin Guillen, Olivier Chevallier, Frédéric Ricolfi, Romaric Loffroy

**Affiliations:** 1Department of Neuroradiology and Emergency Radiology, François-Mitterrand University Hospital, 14 Rue Paul Gaffarel, BP 77908, 21079 Dijon, France; lebeaupin.francois@gmail.com (F.L.); pierre-olivier.comby@chu-dijon.fr (P.-O.C.); marc.lenfant@chu-dijon.fr (M.L.); pierre.thouant@chu-dijon.fr (P.T.); brivael.lemogne@chu-dijon.fr (B.L.); frederic.ricolfi@chu-dijon.fr (F.R.); 2Imaging and Artificial Vision (ImViA) Laboratory-EA 7535, University of Bourgogne/Franche-Comté, 9 Avenue Alain Savary, BP 47870, 21078 Dijon, France; kguillen@hotmail.fr (K.G.); olivier.chevallier@chu-dijon.fr (O.C.); 3Department of Vascular and Interventional Radiology, Image-Guided Therapy Center, François-Mitterrand University Hospital, 14 Rue Paul Gaffarel, BP 77908, 21079 Dijon, France

**Keywords:** intracranial aneurysm, subarachnoid hemorrhage, endovascular treatment, self-expandable stent, coiling, remodeling technique

## Abstract

To assess the efficacy and safety of the Leo stent used alone or with coiling to treat complex intracranial aneurysms (IAs) not eligible for simple or balloon-assisted coiling, this single-center retrospective study included consecutive adults with ruptured or unruptured IAs treated in 2011–2018 by stenting with or without coiling. The indication for stenting was IA complexity precluding simple or balloon-assisted coiling. Extensive data on the patients, IAs, antiplatelet treatments, procedures, and outcomes over the first 36 months were collected. Risk factors for early complications (univariate analysis) and delayed ischemia (multivariate analysis) were sought. We include 64 patients with 66 IAs. The procedural success rate was 65/66 (98.5%). Obliteration was Raymond Roy class I or II for 85% of IAs. Six patients died including four of the 12 patients presenting with subarachnoid hemorrhage, which was the only significant risk factor for early major complications. At 1 month, 45/64 (69%) had no disabilities. No rebleeding was reported. Ischemia was detected by routine MRI in 20 (35%) of the 57 patients with long-term data and was asymptomatic in 14. The stent-within-a-stent configuration was the only independent risk factor for ischemia. The Leo stent used alone or with coils to manage challenging IAs was associated with a high procedural success rate and complete or nearly complete IA obliteration of 85% of IAs. The high frequency of ischemia is ascribable to our use of routine serial MRI. In patients with bleeding, the Leo stent was associated with an excess risk of early, major, intracranial complications, as compared to patients without bleeding. Long-term follow-up was marked by the occurrence of ischemic events in the vascular territory of the stent, mostly silent.

## 1. Introduction

Intracranial aneurysms (IAs) are saccular or fusiform dilations of the walls of the arteries that make up the circle of Willis. These dilations nearly always develop at bifurcations or curve convexities, where blood flow turbulences occur and gradually weaken the arterial wall [1,2]. Intracranial arterial dissection is another, less common, cause of subarachnoid hemorrhage (SAH) of which several forms exist, notably fusiform aneurysms and blood blister-like aneurysms [3,4]. The physiopathology of IA is still unknown; genetics, inflammatory and hemodynamics factors are involved. The local imbalance of wall shear stress caused by blood flow promotes the genesis of IAs, modifying the endothelial function and leading to a chronic inflammation. The pro-inflammatory mediator and the leucocyte wall infiltration led to myo-intimal hyperplasia and a disruption of the internal elastic lamina and the media contributing to growth and rupture of AIs [5,6,7]. In the dissecting aneurysms, an intimal injury creates a pseudolumen with a mural hematoma, then intimal thickening around the pseudolumen led to a fusiform aneurysm [4]. Endovascular treatment (EVT) of IAs occupies a major place in the therapeutic armamentarium as it is associated with less morbidity and mortality compared to surgery. EVT consists chiefly in occluding the aneurysmal sac with platinum coils [8,9,10].

Several techniques have been developed to exclude the aneurysmal sac without altering the angio-architecture of the vessel. One example is balloon-assisted coiling (BAC) to prevent coil protrusion into the vessel [9]. Despite continuous improvements in EVT methods, aneurysm recanalization occurs in up to a third of patients [11,12,13,14]. Moreover, EVT of wide-necked aneurysms is difficult and at times incomplete.

Intracranial stenting has earned a place in the treatment of IAs in recent years. Several studies, including a meta-analysis, demonstrated that both stent-assisted coiling (SAC) and stenting alone were useful for the management of IAs, notably after recanalization [15,16,17,18]. SAC consists of the permanent implantation of a stent in the vessel, across the neck of the aneurysm, to increase the effectiveness of coil packing and to bridge the aneurysmal neck. Several stent positioning techniques can be used depending on the configuration of the aneurysm and habits of the interventional radiology teams [19].

Among existing self-expandable stents, the Leo stent (Balt, Montmorency, France) is produced by braiding individual strands of nitinol to obtain a mesh [20]. Leo is the first recoverable closed-cell stent introduced on the market. Important characteristics include the availability of a broad size range and a strong radial force that ensures good support to the mass of coils. Moreover, the Leo stent has the highest metal coverage among the existing self-expandable stents with flow diverter property [21]. Leo is the first recoverable closed-cell stent introduced on the market. Important characteristics include the availability of a broad size range, strong radial force that ensures good support to the mass of coils. Moreover, the stent Leo have the highest metal coverage among the existing self-expandable stent with flow diverter property [21]. They are able to further enhance aneurysmal thrombosis by diverting blood flow toward the parent vessel and away from the IAs. Importantly, the stent can be re-sheathed, recovered, and repositioned after up to 90% of deployment [22].

The Leo stent has been proven effective and safe for the treatment of IAs. However, information is scant on the tolerance of the Leo stent beyond 36 months [17,23]. Furthermore, few studies have assessed the use of the Leo stent alone or with coiling to treat bleeding due to intracranial artery dissection.

The objective of this study was to assess the short- and long-term efficacy and safety of the Leo stent in patients managed for ruptured IAs (RIAs) or unruptured IAs (UIAs) of any type, including dissection.

## 2. Materials and Methods

### 2.1. Study Design and Patients

This was a single-center retrospective observational cohort study. We identified consecutive patients with RIAs or UIAs treated using a Leo stent with or without coiling at the neurointerventional radiology department of the Dijon University Hospital, Dijon, France, between January 2011 and May 2018. At our department, the first-line treatment of IAs is coiling or BAC, and the study involved only the minority of patients with complex IAs requiring stenting.

We included consecutive patients who were 18 years of age or older at the time of treatment and who had one or more IAs requiring curative or prophylactic treatment by the implantation of a Leo stent with at least 36 months follow-up. Patients lost to follow-up or with missing data were excluded.

The data were collected from the medical files of each patient and from the interventional radiology and hospital’s Picture Archiving and Communication System.

Our ethics committee approved the study and waived the requirement for informed patient consent in compliance with French legislation on retrospective studies of anonymized data.

### 2.2. Patient Management and Endovascular Treatment

Patients with UIAs underwent diagnostic angiography under local anesthesia. Contrast was injected into both carotid arteries and both vertebral arteries. Rotational three-dimensional (3D) images were obtained to evaluate the angioarchitecture of the aneurysm. The optimal treatment strategy was decided during a multidisciplinary meeting then explained to each patient during a visit.

EVT was performed under general anesthesia. Antiplatelet therapy with acetylsalicylic acid (Aspirine; UPSA, Ruel Malmaison, France) 75 mg/day, and clopidogrel (Plavix; Sanofi-Aventis, Longjumeau, France) 75 mg/day, was started 8–10 days before the procedure. After the procedure, aspirin was continued for 6 months and clopidogrel for 3 months. The efficacy of clopidogrel was assessed by a vasodilator stimulated phosphoprotein test; ticagrelor was given instead of clopidogrel to resistant patients and, subsequently, since the end of 2015, to all patients after a change in the recommendation about dual antiplatelet therapy in interventional cardiology, with a concomitant increase in treatment duration to 6 months of dual antiplatelet therapy followed by 6 months of single-drug antiplatelet therapy [24,25].

When stent implantation had not been planned but was deemed necessary during the procedure, i.e., when the patients had not received antiplatelet premedication, a tirofiban infusion was started a few minutes before stent deployment. After 24 h, the tirofiban was replaced by a P2Y12 receptor inhibitor (clopidogrel; or ticagrelor; Brilique, Courbevoie, Astra Zeneca, France).

Patients with RIAs were admitted immediately to the neurotrauma critical care unit. Unenhanced computed tomography (CT) followed by CT-angiography of the circle of Willis established the diagnosis. EVT was performed within 24 h to secure the aneurysm. Given the absence of antiplatelet premedication, a tirofiban infusion was started a few minutes before stent deployment.

EVT was performed under general anesthesia via the transfemoral approach. Selective catheterization of the branches of the aortic arch then of the vertebral arteries or carotid arteries depending on the location of the aneurysm was performed using a short introducer sheath (Terumo, Guyancourt, France) and a 6-French guiding catheter (Envoy DA; Cerenovus, Johnson & Johnson, Miami, FL, USA). A Vasco^®^ microcatheter (Balt, Montmorency, France) with the stent was inserted into the target artery according to the manufacturer’s recommendations. The IA was catheterized before or after stent deployment, depending on the treatment strategy. Isotonic saline was infused into the microcatheter and 1000 milliliter isotonic saline containing 1000 IU of heparin (Panpharma, Luitré-Dompierre, France) and 4 mg of nimodipine (Nimotop; Bayer, Saint-Georges-de-Reneins, France) into the guiding catheter.

Unfractionated heparin (50 IU/kg) with 250 mg of aspirin was injected intravenously a few minutes before stent deployment. In patients with blood-clot formation or without antiplatelet premedication, tirofiban was administered as a slow intravenous infusion over 30 min then for 24 h in a dosage appropriate for each patient’s body weight and renal function. A P2Y12 inhibitor was given orally after the infusion.

Stent diameter and length were chosen based on the angiography findings. The stenting technique varied according to the situation in each patient. In particular, the IA was catheterized before or after stent deployment depending on the clinical situation and angiographic features of the IA. An angiogram was obtained at the end of the procedure in all patients. Immediately after the procedure, the patients were transferred to the neurotrauma critical care unit for monitoring.

Each patient was seen 30 days after the procedure by one of the interventional radiologists (but not necessarily by the one who performed the procedure).

Major complications were defined as death, ischemia, neurological deficit, and intracranial bleeding; and minor complications as poor stent positioning, failure of aneurysm catherization through the stent struts, blood-clot formation without ischemia, and alopecia due to the radiation exposure. 

### 2.3. Long-Term Follow-Up

A MRI was obtained routinely 6 months after the procedure, and the patient was seen with the result. Subsequently, MRI and clinical examination was performed after 3–6 months, 12–18 months, 3 years, 5 years and 10 years. When abnormalities were found, additional MRIs were performed.

A 1.5-T or 3-T MRI machine was used. At the minimum, time-of-flight (TOF) MR-angiography of the circle of Willis and an axial FLAIR sequence were obtained. A diffusion sequence was performed at the first follow-up MRI. In patients with RIAs, an axial T2 echo-gradient sequence was added to the minimal protocol. If the neuroradiologist was unable to reach a conclusion based on these images, an MR-angiogram of the circle of Willis with gadolinium injection was obtained.

In the event of changes in the residual neck or recanalization of the aneurysmal sac, diagnostic angiography was performed and the case of the patient was discussed during a multidisciplinary meeting to determine the best management strategy.

### 2.4. Data Collection

The data were collected from the electronic medical files of each patient, from the neurotrauma critical care unit and hospital discharge summaries, from the interventional radiology and hospital’s Picture Archiving and Communication System, and from the outpatient visit files.

The main patient features including age, sex, cardiovascular risk factors, comorbidities, and history of SAH in the patient or family were recorded. The number, location, and shape of the aneurysms managed by EVT were collected. IAs were classified based on their appearance and on the clinical setting. Saccular aneurysms at bifurcations were classified as small (<10 mm), large (10–24 mm), or giant (≥25 mm). The dome-to-neck ratio was computed as the greatest transverse diameter of the aneurysm over the diameter of the neck, and the neck was categorized as narrow (≤4 mm or dome-to-neck ratio ≥ 1.6) or wide (>4 mm or dome-to-neck ratio < 1.6). Dissecting aneurysms were classified according to Mitzutani as Type I (fusiform dilation of an arterial segment with true dissection; Type II (segmental ectasia), Type III (dolichoectatic dissecting aneurysm), or Type IV (saccular aneurysm or blister-like unrelated to a bifurcation) [3]. We recorded SAH severity graded using the World Federation of Neurosurgical Societies (WFNS) system.

The stenting and coiling techniques were determined based on the interventional radiology report and angiography prints. We recorded the number of coils used, if any. To evaluate the various factors involved in stenting, we recorded the number, diameter, and length of the implanted stents. When more than one stent was implanted, to quantitatively evaluate stent use we computed the sum of the lengths of all implanted stents. This total stent length did not reflect the length of the arterial segment covered by the stents since, in some patients, stents were implanted in series using the telescopic technique and, in others, one stent was placed within another to increase the flow-diversion effect (stent-within-a-stent, SWS). The quality of aneurysm occlusion was assessed based on the Raymond-Roy scale (RRS) applied to the angiography obtained routinely at the end of the procedure [26]. In patients with stenting only or with SWS, the quality of the occlusion was evaluated based on the blood flow decrease within the aneurysm.

For each patient, we recorded the number, type, route, and duration of use of each antiplatelet agent.

Intra-procedural complications and early complications documented during the first follow-up visit on day 30 were collected. Complications were classified as extracranial or intracranial and as with vs. without a clinical impact. The modified Rankin score (mRS) was determined using the data obtained at the 1-month visit. Two categories were distinguished according to the clinical impact of any disabilities, i.e., mRS < 2 or ≥2. The mRS before the procedure was not known.

The MRIs obtained during the subsequent follow-up were used to record the quality and stability of aneurysm exclusion and stent permeability. The presence and/or development of residual ischemic lesions was assessed routinely.

### 2.5. Statistical Analyses

The statistical analyses were conducted using R program version 4.0.5 (Lucent Technologies, Bell Laboratories, Murray Hill, NJ, USA). Qualitative variables were described as absolute numbers and percentages and continuous variables as median (interquartile range) according to distribution.

Qualitative variables were compared using the chi-square or Fisher’s exact test depending on which assumptions were met. Quantitative variables were nonnormally distributed and were therefore compared using the nonparametric Mann-Whitney test. For each test, *p* values below 0.05 were taken to indicate significant differences. The first analysis determined whether the occurrence of intraprocedural or early (≤30 days) complications was associated with SAH, location (anterior vs. posterior), type (dissecting IAs or not) and angioarchitecture (fusiform vs. saccular) of the IAs, type and duration of antiplatelet therapy, and stenting technique. Then, we evaluated whether the development of ischemic lesions seen by MRI during follow-up was associated with dissecting aneurysm, type and duration of dual antiplatelet therapy (with duration transformed into a categorical variable: <6 months vs. ≥6 months), location, type and angioarchitecture of the aneurysms and stenting characteristics (total stenting length, number of stents, and SWS).

We then built two multivariable models to identify factors independently associated with the occurrence of an ischemic event during follow-up. Factors associated with *p* values < 0.20 by univariate analysis were entered into the logistic regression model.

## 3. Results

### 3.1. Patients

We identified 66 patients, with 69 IAs, managed during the study period using a Leo stent with or without coiling. These patients accounted for only 6.6% of patients who received EVT for IAs during the study period. Due to missing data, we excluded two patients with three aneurysms, leaving 64 patients and 66 aneurysms for the study. The long-term median follow-up time for 57 IAs was 79 months (IQR (59–96)). Table 1 reports the main patient features.

### 3.2. Aneurysms

Of the 64 patients, one had mirror IAs of the middle cerebral arteries and another required repeat Leo stenting due to recanalization of an ophthalmic internal carotid artery. Thus, we evaluated 66 aneurysms. Table 2 reports their main features. The 12 patients with SAH were considered critically ill. Their median WFNS severity score was 4 (2.75–5) with 5 as the most common score.

Figure 1 illustrates the locations of the IAs. Most IAs (55/66, 83%) were on the carotid arteries or their branches, and among these, 28/55 (51%) were on the right side.

### 3.3. Short-Term Outcomes

#### 3.3.1. Procedures and Immediate Angiographic Outcomes

The SAC methods used included the stent-jack technique (*n* = 18, 27%) (Figure 2), stenting after coiling (*n* = 14, 21%), coiling through the stent struts (*n* = 10, 15%), the jailing technique (*n* = 4, 6%), and rescue stenting (*n* = 4, 6%). Other methods used were the SWS technique (*n* = 11, 17%) and T stenting (*n* = 4, 6%).

Stent deployment failed for 1 IA, yielding a procedural failure rate of 1.5%. In this case, correct positioning of the stent was not feasible, and simple coiling was not feasible given the fusiform aneurysm morphology and location on the posterior communicating artery. The only available option was horizontal stenting across the circle of Willis. However, the risk/benefit ratio of this option was deemed unfavorable and therapeutic abstention with close monitoring was felt to be the best strategy.This patient did not experience any bleeding events during follow-up.

At the end of the procedure, 44/66 (67%) were RRS 1, 12/66 (18%) were RRS 2, and 9/66 (14 %) were RRS 3. Of these 9 IAs, five were fusiform aneurysms in which a decrease in blood flow was noted at the end of the procedure.

#### 3.3.2. Short-Term Clinical Outcomes

During the visit 30 days after the procedure, the mRS was 0 in 45/64 (70.5%) patients, 1 or 2 in 8/64 (12.5%) patients, and higher than 2 in 11 (17%) patients. Among these 11 patients, six died, including four who were managed during acute SAH.

#### 3.3.3. Complications during and within 30 Days after Endovascular Treatment

Six patients died. Identified causes of death were diffuse treatment-refractory vasospasm (*n* = 3), increased intracranial bleeding (*n* = 1), rupture of a fusiform aneurysm that had been partially secured 2 days earlier (*n* = 1), and intracranial bleeding caused by an injury with the micro-guidewire of a perforating vessel (*n* = 1). Seven (11%) patients had ischemia due to blood-clot formation, and among them two had a discontinuation of antiplatelet therapy. Three (5%) had bleeding: one rebleeding of the aneurysm and two vessel injuries due to the micro-guidewire without clinical relevance. One patient had a neurological deficit due to cerebral edema caused by aneurysmal sac thrombosis requiring corticosteroid therapy.

Transient alopecia occurred in one patient due to the radiation exposure during the procedure. Hematomas developed at the femoral puncture site in three patients; they did not cause compression and required no treatment.

Table 3 reports the factors associated with the occurrence of major intracranial complications during or within 30 days after the procedure. Treatment during acute SAH was the only factor significantly associated with major intracranial complications.

### 3.4. Long-Term Outcomes

#### 3.4.1. Long-Term Angiographic Outcomes

Follow-up data beyond 36 months or more were available for 57 of the 66 IAs, in 55 patients. Among these 57 IAs, 40 (70%) were RRS 1 and nine (16%) were RR2, whereas eight (14%) underwent recanalization including six (10.5%) that required further EVT.

The latest diagnosis of recanalization requiring further EVT was made 36 months after the first stenting procedure. Median time to further EVT was 6 months (5.3–19.5 months). Nine (16%) IAs exhibited an improvement in their Raymond-Roy class. At last follow-up, IA exclusion was satisfactory, i.e., Raymond-Roy class I or II) for 49 (86%) IAs.

#### 3.4.2. Long-Term Complications

The main outcome criterion was ischemia diagnosed by MRI during follow-up. Stent patency was evaluated on the MR angiogram of the Willis circle. Ischemic lesions developed in the stent territory in 20 (36%) patients, of whom 14 were asymptomatic. In eight of the 14 asymptomatic patients, the ischemic lesions were seen on the first MRI obtained after dual antiplatelet agent therapy discontinuation, i.e., 1 year after stenting. Of the remaining six patients, three had an mRS ≥ 2 and three an mRS < 2.

During follow-up, three stents in three patients became occluded. Among them, a patient with early recanalization treated by implantation of a flow diverter 6 months after initial stenting experienced stent occlusion after premature discontinuation of the dual antiplatelet regimen. Another patient had stent occlusion diagnosed after 18 months, with no symptoms of ischemia. The remaining patient experienced a stroke 42 months after initial stenting, with an NIHSS score of 2; at hospital discharge, this patient had an mRSS of 1.

Table 4 reports the results of the univariate analysis carried out to identify factors associated with ischemia during follow-up. Dual antiplatelet duration analyzed as a binary variable (<6 months vs. ≥6 months) was not associated with ischemic complications. A greater total stented length, a higher number of stents, and SWS were significantly associated with ischemia, whereas stent diameter was not.

By multivariate analysis adjusted on total stented length, SWS was significantly associated with ischemia during follow-up, albeit with a very wide 95% confidence interval (OR, 10.8 (1.54–111.0), *p* = 0.025). Because fusiform aneurysms are more likely to cause ischemic lesion, we built another model of multivariate analysis adjusted on the angioarchitecture of the IAs (fusiform vs. saccular), SWS was still significantly associated with ischemia during follow-up (OR, 7.88 (1.82–44.9), *p* < 0.01) and the fusiform IAs was not (OR, 0.853 (0.132–4.43), *p* = 0.86). Of the other variables (number of stents, presence or not of dissecting aneurysm) entered into the model, none was significantly associated with ischemia, including total stented length (OR, 0.985 (0.926–1.050); *p* = 0.62). Table 5 reports the main results of the present study compared to other main similar series from the literature.

## 4. Discussion

In this retrospective study of 66 IAs in 64 patients treated by Leo stenting with or without coiling, the procedure failed for a single aneurysm. In the long term, RRS1 or RRS2 obliteration was achieved in 86% of cases. Of the six patients who died, four were among the 12 patients who presented with SAH. A wide variety of stenting techniques were used. After 1 month, 45 patients were free of disability. In the short term, the only factor significantly associated with major intracranial complications within the first 30 days, by univariate analysis, was presentation with SAH (12 of the 64 patients). The deployment failure rate was 1.5%, which is within the ranges reported with the Leo stent and other stents [16,18,29,30,31,32,33]. There was no bleeding or re-bleeding during the follow-up. In the long term, ischemia visible by MRI developed in a third of patients, of whom 70% were asymptomatic. By multivariate analysis, the only factor that independently and significantly predicted a diagnosis of ischemia during follow-up was use of the SWS technique.

Our 86% rate of Raymond-Roy class I or II occlusion is consistent with earlier reports [16,17,23,34]. This high rate was achieved despite the complexity of the IAs, of which 30% were aneurysms with recanalization after prior treatment and 18% were fusiform. All the included IAs were considered too complex to be managed by simple coiling or SAC. Recanalization requiring retreatment occurred in only 3% of the aneurysms with long-term follow-up, which is comparable to the 5.2% retreatment rate in a meta-analysis of studies of wide-neck aneurysms treated by coiling with or without stenting [18]. A retrospective study of 1325 IAs treated by coiling demonstrated a significantly lower recurrence rate when stents were also used (*n* = 216), although procedure-related mortality was significantly higher with the combined technique (4.6% vs. 1.2%) [35]. Similarly, in a meta-analysis, recurrences were significantly less common with SAC, which was also associated with a significantly higher frequency of improved IA obliteration by progressive thrombosis during follow-up of RIAs and UIAs [36]. Stenting supports the coils, packing them into the IA and preventing their protrusion into the arterial lumen [19,35]. Stents also serve as scaffolds that support the neointima, thereby improving endothelialization of the neck. Moreover, the Leo stent exerts a flow-diversion effect related to its higher pore density compared to laser-cut stents and to its greater metal coverage (14% vs. 5% and 10% for the Enterprise and Neuroform stents, respectively) [37,38]. In our study, nine (16%) IAs exhibit improvement of RRS during the follow-up, Cagnazzo et al. showed modification of the flow of the covered arteries [20]. Pumar et al. reported 75% complete occlusion after treatment of 20 intracranial fusiform aneurysms with stent monotherapy [27]. Juszkat et al. also used the Leo stent in a larger cohort for fusiform aneurysms and underlined the challenges with flow diversion for these aneurysms, in highlighting that stent alone treatment was the best practice and had the lowest complication rate [39]. These data underline the flow diverter property of the Leo stent but highlight the importance of coiling as an adjunct. Then, the stent may change the geometry of the carrier vessel and amend the wall shear stress, which promotes the growth of IAs. The resulting flow diversion through the artery decreases flow turbulence within the IA. In a simulation study, the SWS configuration produced greater flow diversion than did the Pipeline flow-diverter device [29].

Major intraprocedural complications occurred for nearly 30% of the IAs, and 6 (9%) patients died, all of them from major intracranial complications. One reason underlying these findings is the high prevalence of SAH, of 19%. Patients who had SAH were critically ill with a median WFNS score of 4. Nearly two-thirds of patients with SAH experienced major procedure-related intracranial complications, including vasospasm and intracranial hypertension, for which the efficacy of medical and or interventional treatments may be limited. Second, in our department, the first-line treatment for IAs is simple coiling or BAC. During the 7.5-year study period, about 1043 IAs were managed endovascularly (about 120 to 150 each year). The 66 IAs included in our study were the most complex cases, requiring SAC and representing only 6.6% of the total. In a retrospective study, mortality was 11% with Leo stents, which were also associated with a higher rate of thrombosis of 14.8% compared to simple coiling [35]. In a small study of 29 patients (29 IAs) treated with the Leo stent, mortality was 7% [30]. A literature review of stent-assisted coiling with any type of stent found a 19% rate of procedure-related complications and a 2% rate of mortality [31]. The most common procedure-related complication was thromboembolism which affected nearly 10% of patients, compared to 12% in our study. Others have also pointed out the increased risk of thrombus formation with stenting [35].

The need for dual antiplatelet therapy when stenting is used increases the risk of bleeding. The bleeding complication rate in our study was similar to that reported by others and, importantly, was within the range found when coiling was used without stenting [32]. The dual antiplatelet regimen changed during the study period due to a modification of recommendations for interventional cardiology. We found no significant association of the type or duration of antiplatelet therapy with the occurrence of ischemic complications. However, the small sample size provided only limited statistical power for detecting such an association.

The subgroup of patients with aneurysmal dissection including a blood blister such as IAs did not have a higher rate of intraprocedural or periprocedural complications. When treating ruptured blister aneurysms, flow diversion becomes of significant importance as for unruptured aneurysms, with no obvious evidence for different complications profile.

Thus, the use of a Leo stent may be a promising alternative to occlusion of the artery at the dissection site, which is currently the reference standard treatment [33]. The EVT of fusiform IAs was not associated with early major complication or late ischemic complication (OR, 0.853 (0.132–4.43), *p* = 0.86) despite the increased risk of ischemic lesion associated with this kind of IAs.

On the other hand, aside from stent-in-stent thrombosis and related sequelae, Akmangit et al. highlighted the dual stenting technique with the Leo stent and the different variations it can be used in [40]. This layer of nuance may explain the high rates of complications and help highlight cohorts of patients that would be good candidates for the Leo stent. The recognition of a higher complication rate in ruptured situations with Leo stents, even when compared to other flow diverters in the ruptured scenario is essential because it could impact the clinical practice of many interventionalists.

Ischemia has been reported to occur after IA stenting in nearly a third of cases. The higher frequency of ischemic complications in our study compared to others is ascribable to our inclusion of asymptomatic cases detected only on routine follow-up MRIs [17,35]. Most studies included only clinical ischemic events. We believe that silent ischemia should also be considered as it may herald the occurrence of symptomatic events. In addition, ischemia may manifest as nonspecific symptoms such as mood changes, chronic fatigue, or difficulty resuming job-related activities [41,42]. These manifestations may be underestimated by the physical examination while adversely affecting quality of life.

Of the 20 patients with ischemic complications in our study, 14 had no symptoms and six had neurological deficits. In our study, most ischemic complications occur shortly after decreasing the intensity of antiplatelet therapy (from 2 to 1 or from 1 to no drugs). Transient ischemic attacks may herald the development of an ischemic lesion, as was the case in two of our patients. Several suggestions have been made to decrease the frequency of ischemia by better selecting patients who are ready for a decrease in antiplatelet therapy intensity. One possibility is to wait until digital subtraction angiography after 8 months demonstrates the absence of intrastent stenosis before switching from dual to single antiplatelet therapy [17]. However, adjacent coils may hinder the evaluation of the stent lumen.

Our analysis of long-term data suggests that the amount of material used may increase the risk of thromboembolism. By univariate analysis, total stented length, number of stents, and SWS were significantly associated with the development of ischemia during follow-up. SWS provides an excellent flow-diverting effect by doubling the covered surface area, whereas the deployment of two or more stents using the telescopic technique allows the coverage of a longer arterial segment, thereby obviating the need for long stents that are more difficult to navigate within the circle of Willis arteries [29]. In the multivariate analysis adjusted for total stented length, only SWS was significantly associated with the occurrence of ischemic events during follow-up. The length of stenting was a cofounding factor. This finding raises the issue of the endothelialization of stents placed inside each other. In an interventional study, stent overlapping resulted in increased stent thickness, diminished local blood flow with decreases in shear stress and blood flow, and a longer time to endothelialization resulted in a higher incidence of intrastent stenosis [43]. A retrospective review of 123 IAs treated by SAC found that having three or four factors among hypertension, diabetes, dyslipidemia, and smoking was non-significantly associated with intrastent stenosis by univariate analysis (31.8% vs. 12.9%, *p* = 0.05) [44]. We did not replicate this finding, probably due to missing data about whether smoking was past or current and to the small sample size. A better analysis of cardiovascular risk factors might help to select patients for Leo stenting.

The use of flow diversion stents such as the Pipeline Embolization Device (PED; Covidien, Irvine, CA, USA) or Silk Flow Diverter (Balt Extrusion, Montmorency, France) to treat distal and ruptured IAs is growing. Two meta-analyses showed an effective aneurysmal occlusion but a higher rate of complications, mostly ischemic and thrombo-embolic [45,46]. Moreover, the low-profile intracranial stents such as the Leo baby stent can be deployed into thin arteries and delivered through microcatheters (with an internal diameter of 0.0165 inches) which allows easier navigation in the vessel. Thus, a mid flow-redirecting stent could be a good balance between coiling and stenting with flow diverters in the endovascular treatment of distal intracranial aneurysms or located in the posterior circulation.

Our study had some limitations. The main limitation is its retrospective design, with the only available data being those recorded in the medical files and, probably, some uncertainties due to the lack of standardization. Furthermore, our population was heterogeneous, with various causes of IA and widely differing clinical presentations, particularly regarding the presence of SAH. These various forms of IA differ in their prognosis and are not readily comparable. Nevertheless, our pragmatic study provides a view of real-life practice in our vascular interventional radiology department.

## 5. Conclusions

SAC using the Leo stent provided a high rate of obliteration of complex IAs, irrespective of the clinical presentation, by virtue of an increase in coiling density, and of the flow-diverting properties of the Leo stent although moderate. The obliteration was durable in the long term. The Leo stent offers an alternative to artery occlusion for the treatment of dissecting IAs. When performed in patients with RAIs versus UAIs, EVT using the Leo stent is associated with an excess risk of intraprocedural and periprocedural complications. In these cases, stenting should be considered as a last resort.

Ischemia within the stented territory was common during follow-up but usually asymptomatic and coincidental with changes in the antiplatelet therapy regimen. Twelve months dual and then lifelong single antiplatelet therapy, as in interventional cardiology, could help reduce occurrence of ischemia. SWS is independently associated with an increased risk of ischemia during follow-up compared to other techniques. Thus, the stenting indication and technique should be planned as accurately as possible. Further work is needed to validate our findings, notably multicenter randomized trials.

## Figures and Tables

**Figure 1 jcm-10-04541-f001:**
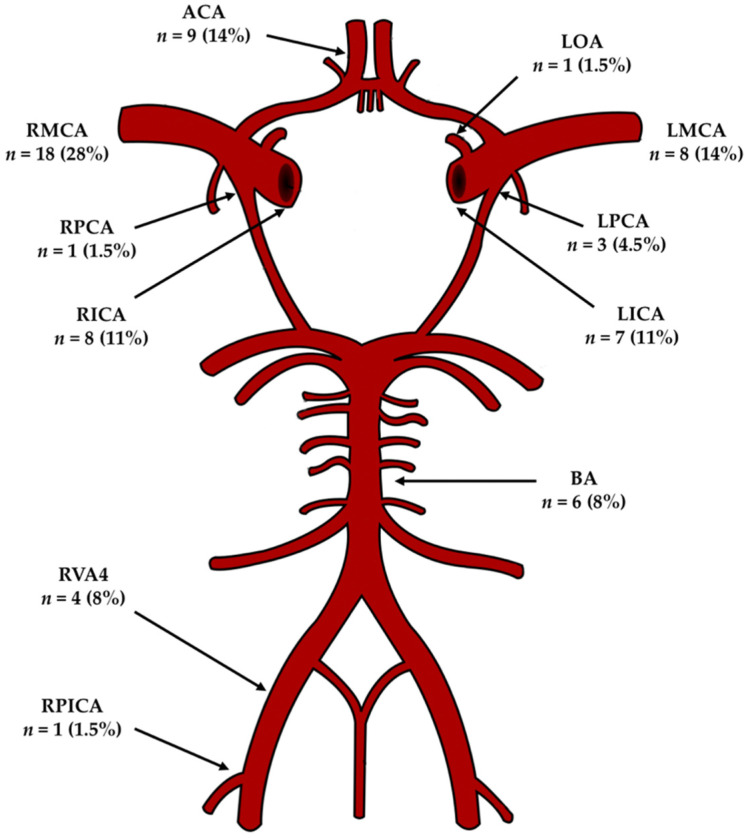
Diagram showing the locations of the 66 aneurysms on the circle of Willis. ACA, anterior communicating artery; LOA, left ophthalmic artery; LMCA, left middle cerebral artery; LPCA, left posterior communicating artery; LICA, left internal carotid artery; BA, basilar artery; RPICA, right posterior inferior cerebellar artery; RVA4, segment V4 of the right vertebral artery; RICA, right internal carotid artery; RPCA, right posterior communicating artery; RMCA, right middle cerebral artery.

**Figure 2 jcm-10-04541-f002:**
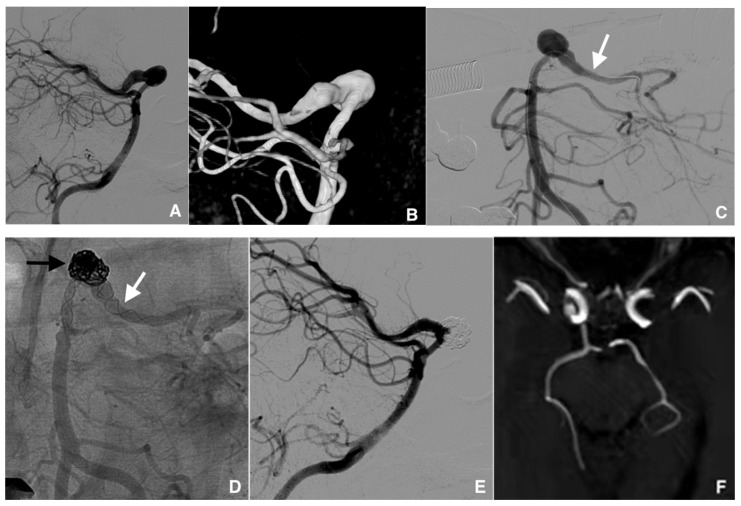
Example of management of a dissecting intracranial aneurysm (IA) (length, 20 mm; wide, 7 mm) in a 52-year-old woman treated with stent-jack technique. (**A**) Pre-treatment digital subtraction angiography (DSA) of the fusiform IA of the left posterior communicating artery (lateral view). (**B**) Image reconstruction of the IA with volume rendering technique. (**C**) Cathetherism of the aneurysm-carrying vessel (white arrow) in a posterior–anterior DSA projection. (**D**) Nonsubtracted post-treatment control DSA image, posterior-anterior projection, showing the widest part of the IA filled with coils (black arrow) and two Leo stents implanted using the telescopic technique (white arrow). (**E**) DSA image at the end of the procedure, lateral projection: the IA is partially excluded from the intracranial circulation. (**F**) Magnetic resonance imaging with 3D time-of-flight view, at 36 months posttreatment control: the IA is completely excluded and the formerly aneurysm-carrying vessel displays regular endoluminal signal.

**Table 1 jcm-10-04541-t001:** Main features of the 64 patients.

Features	Data
Males/Females, *n* (%)	20 (31)/44 (69)
Age at stenting, years, median (IQR) (range)	53 (47–62) (22–74)
Cardiovascular risk factors, *n* (%)	
*Smoking (past or current)*	26 (41)
*Hypertension*	35 (55)
Comorbidities, *n* (%)	
*Prior SAH*	17 (26.5)
*Prior SAH in family member*	7 (11)
*Hepatorenal polycystic disease*	3 (5)
*Fibromuscular dysplasia*	1 (1.6)
Number of IAs, *n* (%)	
*1*	43 (68)
*≥ 1*	21 (33)
*2*	14 (22)
*3*	7 (11)

SAH, subarachnoid hemorrhage; IA, intracranial aneurysm; IQR, interquartile range.

**Table 2 jcm-10-04541-t002:** Clinical presentation and main features of the 66 intracranial aneurysms in 64 patients managed with one or more Leo stents.

Features	*n* = 66 Aneurysms (64 Patients)
**Clinical presentation**	
Asymptomatic, *n* (%)	23 (35)
Recanalization, *n* (%)	20 (30)
SAH, *n* (%)	12 (19)
Hemorrhagic dissection, *n* (%)	10 (15)
Saccular IA rupture, *n* (%)	2 (3)
Headache without SAH, *n* (%)	6 (9)
Dissecting IA, *n* (%)	5 (7.8)
Unexplained, *n* (%)	1 (1.5)
Compression, *n* (%)	4 (6)
Ischemia, *n* (%)	1 (1.5)
**Features of the IAs**	
Morphology, *n* (%)	
*Saccular (bifurcation or sidewall)*	43 (65)
*Dissecting aneurysm*	23 (35)
*Mitzutani type*	
*I*	11 (48)
*II*	3 (13)
*III*	2 (9)
*IV*	7 (30)
Size, mm, *n* (%)	
*<10*	47 (73)
*10 to <25*	17 (26)
*≥25*	2 (3.2)
Neck, *n* (%)	
*Size ≥ 4 mm*	18 (33)
*Dome-to-neck ratio ≤ 1.6*	46 (87)
*Fusiform*	12 (17)

SAH, subarachnoid hemorrhage; IA, intracranial aneurysm.

**Table 3 jcm-10-04541-t003:** Univariate analysis to identify factor associated with major intracranial complication immediately after or within 30 days after stenting.

Variables	No Complication Group *n* = 48 (73%)	Complication Group *n* = 18 (27%)	Total *n* = 66 (100%)	Crude OR (IC 95%)	*p* Value
**Clinical presentation, *n* (%)**					
SAH					
*No*	43 (90)	11 (61)	54 (82)	1	-
*Yes*	5 (10)	7 (39)	12 (18)	5.3(1.19; 25.76)	0.013
Dissecting aneurysm					
*No*	32 (67)	11 (61)	43 (65)	1	-
*Yes*	16 (33)	7 (39)	23 (35)	1.27 (0.40; 3.88)	0.67
Antiplatelet agents					
*Clopidogrel*	37 (77)	9 (56)	46 (70)	1	-
*Ticagrelor*	11 (23)	7 (44)	18 (30)	2.62 (0.78; 8.75)	0.12
Location					
*Anterior circulation*	42 (88)	13 (72)	55 (83)	1	-
*Posterior circulation*	6 (12)	5 (28)	11 (17)	2.69 (0.68; 10.4)	0.15
**IA characteristics *n* (%)**					
Size, mm					
*<10*	37 (75)	11 (72)	49 (74)	1	-
*≥10*	12 (25)	5 (28)	17 (26)	1.15 (0.32; 3.80)	0.82
Angioarchitecture					
*Saccular*	39 (81)	16 (89)	55 (83)	1	-
*Fusiform*	9 (19)	3 (17)	11 (17)	0.87 (0.18; 3.38)	0.85
Blister like IA					
*No*	45 (94)	14 (78)	59 (89)	1	
*Yes*	3 (6)	4 (22)	7 (11)	4.29 (0.85; 24.0)	0.077
**Stenting technique, *n* (%)**					
Number of stents					
*1*	32 (67)	13 (72)	45 (68)	1	
*>1*	16 (33)	5 (28)	21 (32)	0.77 (0.22; 2.45)	0.67
Rescue stenting					
*No*	45 (94)	16 (89)	61 (92)		-
*Yes*	3 (6)	2 (11)	5 (8)	1.86 (0.14; 17.79)	0.61
Stenting for recanalization					
*No*	28 (58)	14 (78)	42 (64)	1	-
*Yes*	20 (42)	4 (22)	24 (36)	0.40 (0.10; 1.31)	0.14
Stent within stent					
*No*	37 (77)	15 (83)	52 (79)	1	-
*Yes*	11 (23)	3 (17)	14 (21)	0.67 (0.14; 2.53)	0.58

SAH, subarachnoid hemorrhage; IA, intracranial aneurysm; OR, odds ratio; IC, interval confidence.

**Table 4 jcm-10-04541-t004:** Univariate analysis to identify factors associated with ischemia during follow-up.

	No Ischemia Group *N* = 37 (65%)	Ischemia Group *N* = 20 (35%)	Total *N* = 57 (100%)	Crude OR (IC 95%)	*p* Value
**Variables**	53.0 (47.0; 61.0)	53.5 (43.5; 58.8)	57	0.995 (0.950; 1.04)	0.82
**Clinical setting at stenting, *n* (%)**					
*No*	29 (78)	12 (60)	41 (72)	1	-
*Yes*	8 (22)	8 (40)	16 (28)	2.42 (0.734; 8.11)	0.15
Antiplatelet agents					
*Clopidogrel*	24 (65)	16 (80)	40 (70)	1	-
*Ticagrelor*	13 (35)	4 (20)	17 (30)	0.462 (0.114; 1.58)	0.24
Curent or former smocker					
*No*	21 (58)	10 (53)	32 (56)	1	-
*Yes*	15 (42)	9 (47)	25 (44)	1.26 (0.408; 3.89)	0.69
Location					
*Anterior circulation*	31 (84)	19(95)	50 (88)	1	-
*Posterior circulation*	6 (12)	1 (5)	7 (12)	2.69 (0.68; 10.4)	0.15
**IA characteristics *n* (%)**					
Size, mm					
*<10*	28 (76)	15 (75)	43 (75)	1	-
*≥10*	9 (24)	5 (25)	14 (26)	1.04 (0.277; 3.59)	0.95
Angioarchitecture					
*Saccular*	32 (86)	16 (80)	48 (84)	1	-
*Fusiform*	5 (14)	4 (20)	9 (16)	1.60 (0.354; 6.87)	0.52
Blister like IA					
*No*	36 (97)	17 (85)	53 (93)	1	
*Yes*	1 (3)	3 (15)	4 (7)	6.2 (0.45; 34.1)	0.12
**Stenting technique, *n* (%)**					
Number of stents					
*1*	30 (81)	9 (45)	39 (68)	1	<0.01
*>1*	7 (19)	11 (55)	18 (32)	5.24 (1.61; 18.4)	
Total stented length, mm, median (IQR)	18 (18–25)	30 (18–37.8)	57 (100)		0.012
Stent diameter, mm, median (IQR)	2.5 (2.5–3.5)	2.75 (2.5–3.5)	57 (100)		0.350
Stenting for recanalization					
*No*	23 (62)	11 (55)	34 (60)	1	-
*Yes*	14 (38)	9 (45)	23 (40)	1.34(0.44; 4.08)	0.6
Stent within a stent					
*No*	34 (92)	12 (60)	46 (81)	1	-
*Yes*	3 (8)	8 (40)	11 (19)	7.56 (1.86; 39.1)	<0.01

IA, intracranial aneurysm; IQR, interquartile range.

**Table 5 jcm-10-04541-t005:** Main outcomes of the main series from the literature using Leo stent for treatment of intracranial aneurysms.

Variables	Our Study 2021	Voigt et al., 2017 [16]	Sedat et al., 2017 [17]	Lubicz et al., 2017 [23]	Pumar et al., 2013 [27]	Lv et al., 2011 [28]
Patients (*n*)	64	39	153	48	20	28
IAs (*n*)	66	40	155	50	20	28
*Saccular*	54	-	153	50	0	27
*Fusiform*	12	-	2	0	20	1
Ruptured IAs (%)	12	7	0	6	3	0
Stenting failure (%)	1.5	0	1.9	3.8	0	7
Mortality (%)	9	0	0	0	0	0
Complications (%)						
*Immediate*	27	15	19	2	15	11
*Delayed*	10	-	2	4	0	0
Morbidity (%)	17	-	9.1	0	15	0
Ischemic lesion during fu * (%)	35	-	-	-	-	-
Occlusion rate (%)						
*Immediate*	67	-	73	76	-	71
*Delayed*	86	89	97.5	70	75	86

IA, intracranial aneurysm; *n*, number; fu, follow-up; * with magnetic resonance imaging; -, not available.

## Data Availability

All the study data are reported in this article.

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
