# Peer review of "Short- and Long-Term Safety and Efficacy of Self-Expandable Leo Stents Used Alone or with Coiling for Ruptured and Unruptured Intracranial Aneurysms: A Retrospective Observational Study"

_jcm, 2021, doi:10.3390/jcm10194541_

Round 1

Reviewer 1 Report

Dr. Lebeaupin and colleagues present a clinical article where they summarize the data of their overall experience about the use of the Leo stent, with or without coiling, in the management of intracranial aneurysms. 

Leo is a well know, relatively old, self-expanding, nitinol stent generally used for stent-assisted coiling of broad-necked and fusiform intracranial aneurysms. 

Because of the later implementation of the pipeline embolization device, with a long series of further flow diverters deriving from it, Leo has progressively lost interest to the point that the topic of this article risks being anachronistic and of less importance for the readers.    

Introduction

It should be more focused on the properties of the Leo stent and its aspects are still unknown, the reason for which this study has been conducted. All the aspects regarding the incidence and prognosis of subarachnoid hemorrhage and other endovascular techniques are beyond the interest of this article. It would be better instead to briefly condense the main facts regarding the natural history of broad-based, large, and giant aneurysms for which Leo stent has indication. For your convenience read and cite the following references:   

  • Juvela S, Poussa K, Lehto H, Porras M. Natural history of unruptured intracranial aneurysms: a long-term follow-up study. Stroke. 2013 Sep;44(9):2414-21. doi: 10.1161/STROKEAHA.113.001838. Epub 2013 Jul 18. PMID: 23868274.
  • Du Boulay GH. Some observations on the natural history of intracranial aneurysms. Br J Radiol. 1965 Oct;38(454):721-57. doi: 10.1259/0007-1285-38-454-721. PMID: 5828573.
  • Giotta Lucifero A, Baldoncini M, Bruno N, Galzio R, Hernesniemi J, Luzzi S. Shedding the Light on the Natural History of Intracranial Aneurysms: An Updated Overview. Medicina (Kaunas). 2021 Jul 22;57(8):742. doi: 10.3390/medicina57080742. PMID: 34440948; PMCID: PMC8400479.

Methods 

Study Design represents the main weakness of the article. The authors present no data about the type, location, and size of the treated aneurysms, this point compromising the interpretation of the results. My suggestion is to stratify at least the data per class, e.g., anterior vs. posterior circulation, type of parental vessel, e.g. (ICA, VA, BA, etc.), and angioarchitecture (e.g., saccular, fusiform, large, giant, etc.). Blood-blister-like are known to have a different natural history and pathophysiology and they should be excluded to avoid a bias. The same concept is valid also for dissecting aneurysms.  

Results

Figure 1 is unnecessary. 

“Anatomical Outcomes” is meaningless. They probably intend angiographic outcomes. 

Avoid repetitions. Raymond-Roy classification has been assumed as the outcome measure for angiographic outcome already in Methods. It is unnecessary to repeat this point in Results. 

The illustrative case should be postponed at the end of the Results. 

Table 3 has no sense. It should report the independent predictors of ischemia or other major complications based on the uni- and multivariate analysis.

Table 4 regarding the univariate analysis does not report the OR and CI 95%. 

I suggest adding a table aimed to briefly compare the main results of the present study (aneurysm exclusion, outcome, and complication rate) with other similar series reported in the literature. 

Discussion

The discussion is too long and dispersive. Try to reduce its length and stay focused on the clinical results of the stent and comparison with similar studies already reported.  

Discuss the main differences, pros, and cons compared to the flow diverters. 

Conclusion

By the reason of the aforementioned wide series of biases, the conclusions cannot be supported by the results.    

The article needs a substantial revision starting from the design. 

A point of interest about this stent regards its reported moderately flow-redirecting properties. I would analyze these aspects in the Discussion, making them more interesting and actual the article.   

Author Response

Responses to Reviewer 1 Comments

Thank you very much for your relevant comments. Please find below our replies.

Dr. Lebeaupin and colleagues present a clinical article where they summarize the data of their overall experience about the use of the Leo stent, with or without coiling, in the management of intracranial aneurysms. 

Leo is a well know, relatively old, self-expanding, nitinol stent generally used for stent-assisted coiling of broad-necked and fusiform intracranial aneurysms. 

Because of the later implementation of the pipeline embolization device, with a long series of further flow diverters deriving from it, Leo has progressively lost interest to the point that the topic of this article risks being anachronistic and of less importance for the readers.    

Reply: Thank you very much for your comments. We fully agree with you about the new stents currently available on the market. However, the Leo stent is still used and it is always valuable to evaluate properly technologies in clinical practice before using new other devices. That’s what we did here with a great number of patients.

Introduction

It should be more focused on the properties of the Leo stent and its aspects are still unknown, the reason for which this study has been conducted. All the aspects regarding the incidence and prognosis of subarachnoid hemorrhage and other endovascular techniques are beyond the interest of this article. It would be better instead to briefly condense the main facts regarding the natural history of broad-based, large, and giant aneurysms for which Leo stent has indication. For your convenience read and cite the following references:   

  • Juvela S, Poussa K, Lehto H, Porras M. Natural history of unruptured intracranial aneurysms: a long-term follow-up study. Stroke. 2013 Sep;44(9):2414-21. doi: 10.1161/STROKEAHA.113.001838. Epub 2013 Jul 18. PMID: 23868274.
  • Du Boulay GH. Some observations on the natural history of intracranial aneurysms. Br J Radiol. 1965 Oct;38(454):721-57. doi: 10.1259/0007-1285-38-454-721. PMID: 5828573.
  • Giotta Lucifero A, Baldoncini M, Bruno N, Galzio R, Hernesniemi J, Luzzi S. Shedding the Light on the Natural History of Intracranial Aneurysms: An Updated Overview. Medicina (Kaunas). 2021 Jul 22;57(8):742. doi: 10.3390/medicina57080742. PMID: 34440948; PMCID: PMC8400479.

Reply: Thank you for your comments. As suggested, the paper has been improved to be more focused on the properties of the Leo stent, the reason for which this study has been conducted, both in the introduction and the discussion sections. All the aspects regarding the incidence and prognosis of subarachnoid hemorrhage and other endovascular techniques in the introduction especially have been removed. Instead, new paragraphs have been added in the introduction section to describe the natural history of broad-based, large, and giant aneurysms, and the properties of the Leo stent. The 3 proposed references have been cited as suggested. The references have been renumbered accordingly.

Methods 

Study Design represents the main weakness of the article. The authors present no data about the type, location, and size of the treated aneurysms, this point compromising the interpretation of the results.

Reply: Thank you very much for your comments. As suggested, the main data about the population, the clinical presentation, and the main features including the shape/angioarchitecture of the aneurysm, the size, the location and the type of dissecting aneurysm according to the Mizutani classification have been added and are now present in the tables 1 and 2. Moreover, the figure 2 is a diagram showing the location of the 66 IAs. More details are provided in both tables 3 and 4 too.

My suggestion is to stratify at least the data per class, e.g., anterior vs. posterior circulation, type of parental vessel, e.g. (ICA, VA, BA, etc.), and angioarchitecture (e.g., saccular, fusiform, large, giant, etc.). Blood-blister-like are known to have a different natural history and pathophysiology and they should be excluded to avoid a bias. The same concept is valid also for dissecting aneurysms.  

Reply: Thank you very much for your comments. We have added in the tables 3 and 4 the different subgroups to stratify the data per class and make them more visible, as suggested. Subgroups analyses have been performed. However, we have chosen to leave the Blood blister like aneurysms and dissecting aneurysms, even if the natural history and physiopathology are different because this specific point is still controversial. In our mind, the main objective of this work was the retrospective evaluation of the Leo stent in its entirety in order to have a pragmatic vision of its effectiveness and its drawbacks whatever the situation.

Results

Figure 1 is unnecessary. 

Reply: Thank you very much for your comments. Figure 1 has been removed as suggested. All other figures have been renumbered.

“Anatomical Outcomes” is meaningless. They probably intend angiographic outcomes. 

Reply: Thank you very much for your comments. The term “Anatomical” has been changed for “Angiographic” as suggested.

Avoid repetitions. Raymond-Roy classification has been assumed as the outcome measure for angiographic outcome already in Methods. It is unnecessary to repeat this point in Results. 

Reply: Thank you very much for your comments. This repetition has been corrected as suggested.

The illustrative case should be postponed at the end of the Results. 

Reply: Thank you very much for your comments. The illustrative case has been postponed at the end of the results section as suggested.

Table 3 has no sense. It should report the independent predictors of ischemia or other major complications based on the uni- and multivariate analysis.

Reply: Thank you very much for your comments. The table 3 has bene improved as suggested. We added the crude OR to identify factors associated with an early major complication in univariate analysis. We added in the table 3 the different subgroups, including the location of the aneurysm, the angioarchitecture (fusiform vs sacciform), and the size of the IAs to flesh out the table.

Table 4 regarding the univariate analysis does not report the OR and CI 95%. 

Reply: Thank you very much for your comments. This repetition has been corrected as suggested. The table 4 has been improved as suggested. We added some subgroups and OR with CI 95% as suggested.

I suggest adding a table aimed to briefly compare the main results of the present study (aneurysm exclusion, outcome, and complication rate) with other similar series reported in the literature. 

Reply: Thank you very much for your comments. As suggested, a new table (table 5) has been added at the end of the result section comparing the main results of the present study to other main similar series from the literature.

Discussion

The discussion is too long and dispersive. Try to reduce its length and stay focused on the clinical results of the stent and comparison with similar studies already reported.  

Reply: Thank you very much for your comments. The discussion has been shorten as suggested. The paragraph about MR angiography has bene deleted to focus on clinical results and comparison with other studies, as suggested.

Discuss the main differences, pros, and cons compared to the flow diverters. 

Reply: Thank you very much for your comments. A suggested, we discussed the pros and cons to the flow diverters in a specific paragraph below at the end of the discussion section: “The use of flow diversion stents such as Pipeline Embolization Device (PED; Covidien, Irvine, California) or Silk flow diverter (Balt Extrusion, Montmorency, France) to treat distal and ruptured IAs is growing. Two meta-analyses showed an effective aneurysmal occlusion but a higher rate of complications, mostly ischemic and thrombo-embolic [42, 43]. Moreover, the low-profile intracranial stents such as the Leo baby stent can be deployed into thin arteries and delivered through microcatheters (with an internal diameter of 0.0165 inches) which allows and easier navigation in the vessel. Thus, mid flow-redirecting stent could be a good balance between coiling and stenting with flow diverters in the endovascular treatment of distal intracranial aneurysms or located in the posterior circulation.” Two more references have been added (the last 2 ones).

Conclusion

By the reason of the aforementioned wide series of biases, the conclusions cannot be supported by the results.    

The article needs a substantial revision starting from the design. 

A point of interest about this stent regards its reported moderately flow-redirecting properties. I would analyzethese aspects in the Discussion, making them more interesting and actual the article.  

Reply: Thank you very much for your comments. The conclusion has been modified. The article has been substantially improved as suggested. The point of interest about this Leo stent regarding its reported moderately flow-redirecting properties has been highlighted in the discussion section.

Author Response

Responses to Reviewer 2 Comments

Thank you very much for your relevant comments. Please find below our replies.

The authors write a paper entitled, “Short- and long-term safety and efficacy of self-expandable Leo stents used alone or with coiling for ruptured and unruptured intracranial aneurysms: a retrospective observational study.” The authors have compiled a retrospective review of their cases from what appear to be a single institution from the years 2011-2018. They have follow-up for their patients that appears to range from 6 months to 10 years, although this is likely limited because of the date of submission to this journal. I commend the authors for their description and analysis of their cases in 64 patients with 66 aneurysms. They compartmentalized treatment plans based on types of aneurysms and rupture status. This being said, the paper attempts to demonstrate the risks and potential uses of treating aneurysms with simple flow diversion properties of Leo stents, stent assisted coiling cases, and telescoping stent cases for aneurysms that have a broad range from dissecting aneurysms, ruptured saccular aneurysms, and fusiform aneurysms.

Reply: Thank you very much for your comments. Indeed, we included patient with at least a 36-month follow-up (median follow-up time was 79 months with IQR [59-96]).

The authors include multiple forms of treatment as well as multiple morphologies of treatment in this one paper and look at short and long terms outcomes including mortality and morbidity. They had a complication rate that was near 27% based on Table 3. I commend the authors for creating Figure 2 and Table 2 as it gives an initial and good breakdown of the different morphologies of aneurysm that were treated. Based on Page 4, section 2.4, lines 169-175, I think that this paper has real potential if the authors were to identify the different subgroups of aneurysms and complications related to each. For example, the authors highlight the flow diversion capabilities of the Leo stent in the discussion and how there is a higher rate of ischemia with stent-in-stent techniques, but this is likely to happen for fusiform aneurysms and this is not clear when these circumstances arose. I do believe that this paper would be stronger if aneurysms were split according into fusiform, saccular, and blister aneurysms and each subdivided into ruptured and unruptured. This can be done in place of or in conjunction with the Mitzutani classification. This is important because highlighting flow diversion versus adjunct treatment options with the Leo stent becomes more obvious and may show the results in a different light and help actually shape clinical management.

Reply: Thank you very much for your comments. As suggested we improved the paper by adding in the tables 3 and 4 different subgroups to stratify the data according to location of IAs (anterior vs posterior), angioarchitecture (fusiform vs saccular), type of IAs (Blister Blood like IAs vs not). The first analysis (short-term) did not show any association between the different sub-groups previously mentioned and major complications. The only factor associated with a major complication was the acute SAH. The second analysis (long-term) did not show any association between occurrence of ischemia in the stent territory (anterior vs posterior) and fusiform aneurysm. Because fusiform aneurysms are more likely to cause ischemic lesion, we built a second model of multivariate analysis adjusted on the angioarchitecture of the IAs, SWS was still significantly associated with ischemia during follow-up (OR 7.88 [1.82; 44.9], p< 0.01) and the fusiform IAs was not (OR 0.853 [0.132; 4.43], p=0.86).

In addition, the discussion of the treatment of flow diversion qualities of Leo stents in fusiform aneurysms should also highlight some of the earlier work that is shown below:

Juszkat R, Nowak S, Smól S, Kociemba W, Blok T, Zarzecka A. Leo stent for endovascular treatment of broad-necked and fusiform intracranial aneurysms. Interv Neuroradiol. 2007 Sep;13(3):255-69. doi: 10.1177/159101990701300305. Epub 2007 Sep 15. PMID: 20566117; PMCID: PMC3345341.

Pumar JM, Arias-Rivas S, Rodríguez-Yáñez M, Blanco M, Ageitos M, Vazquez-Herrero F, Castiñeira-Mourenza JA, Masso A. Using Leo Plus stent as flow diverter and endoluminal remodeling in endovascular treatment of intracranial fusiform aneurysms. J Neurointerv Surg. 2013 Nov;5 Suppl 3:iii22-7. doi: 10.1136/neurintsurg-2013-010661. Epub 2013 Apr 12. PMID: 23585639.

Each of these series also used the Leo stent in larger cohorts for fusiform aneurysms and highlighted the challenges with flow diversion for these aneurysms. These papers highlighted that stent alone treatment was the best practice and had the lowest complication rate and highlighted the importance of coiling as an adjunct. This is not clear from the current paper and would be nice to highlight.

Reply: Thank you very much for your comments. These 2 studies have been added in the reference list (references 32 and 33) and discussed in the discussion section as suggested, to highlight the challenges with flow diversion for fusiform aneurysms. All references have been renumbered accordingly.

Similarly, there are a high proportion of blister aneurysms treated in this study with 35% of patients having blister aneurysms. When treating ruptured blister aneurysms, flow diversion becomes of significant import and this is the same for unruptured aneurysms, but understanding the difference between the complication profile for these authors would be something that would add to the literature and is needed in this paper.

Reply: Thank you very much for your comments.However, in this study 23 (35%) were dissecting IAs according to the Mitzutani classification. Among them, only 7 (11%) were blood blister like aneurysms (cf. 3.2 Aneurysms, table 2). We did not find any differences in the complication profile for the IAs beyond their “ruptured or not” status, as in our study.  The subgroup of patients with aneurysmal dissection including the blood blister like IAs did not have a higher rate of intraprocedural or periprocedural complications in our study. Then the following sentence has been added in the discussion section: “When treating ruptured blister aneurysms, flow diversion becomes of significant importance as for unruptured aneurysms, with no obvious evidence for different complications profile”.

Finally, the authors highlight that size and need for multiple stents had higher rates of periprocedural complications. Again, aside from stent-in-stent thrombosis and related sequelae, Akmangit et al highlights the dual stenting technique with the Leo stent and the different variations it can be used in. This layer of nuance is needed to actually explain the high rates of complication in this paper and would also help highlight cohorts of patients that would be good candidates for the Leo stent. The above paper citation is included below:

Akmangit I, Aydin K, Sencer S, Topcuoglu OM, Topcuoglu ED, Daglioglu E, Barburoglu M, Arat A. Dual stenting using low-profile LEO baby stents for the endovascular management of challenging intracranial aneurysms. AJNR Am J Neuroradiol. 2015 Feb;36(2):323-9. doi: 10.3174/ajnr.A4106. Epub 2014 Sep 18. PMID: 25234031; PMCID: PMC7965656.

This paper overall is a very good discussion of the recognition of a higher complication rate in ruptured situations with Leo stents, even when compared to other flow diverters in the ruptured scenario. In addition, they add to the literature highlighting a higher complication rate in different populations that could impact the clinical practice of many interventionalists.

Reply: Thank you very much for your comments. All these features have been discussed in the discussion section and this reference has been added. All references have been renumbered consequently.

Things to make this paper stronger:

- Include 3 citations included above and help that shape the discussion.

Reply: Thank you very much for your comments. The 3 above citations have been added in the reference list and discussed, as suggested.

- Include in the discussion the high complication rate based on the different morphologies and rupture status that help delineate good potential patients who would benefit from Leo stents

Reply: Thank you very much for your comments. As suggested, we added this point in the discussion section. Unfortunately, we did not succeed to highlight any association between the size of the AIs, or shape. We have already discussed about the rupture status.

- Include discussion of fusiform and blister aneurysms separately and the need for flow diversion qualities associated with Leo stents

Reply: Thank you very much for your comments. This point has been added in the discussion section as suggested.

- High rate of blister aneurysms and understanding the healing property and long-term follow-up (ie endothelialization) would be helpful in the discussion.

Reply: Thank you very much for your comments. However, the rate of blood blister like IAs was not so high in our study since they account for 7 (11%), 23 (35%) IAs corresponding to all dissecting aneurysms. We made it more understandable in the table 2. Moreover, we added this subgroup in the two analyses, for allowing better understanding of the different types of IAs.